# LABEL AUGMENTATION WITH REINFORCED LABELING FOR WEAK SUPERVISION

## ABSTRACT

Weak supervision (WS) is an alternative to the traditional supervised learning to address the need for ground truth. Data programming is a practical WS approach that allows programmatic labeling data samples using *labeling functions* (LFs) instead of hand-labeling each data point. However, the existing approach fails to fully exploit the domain knowledge encoded into LFs, especially when the LFs' coverage is low. This is due to the common data programming pipeline that neglects to utilize data features during the generative process. This paper proposes a new approach called *reinforced labeling* (RL). Given an unlabeled dataset and a set of LFs, RL augments the LFs' outputs to cases not covered by LFs based on similarities among samples. Thus, RL can lead to higher labeling coverage for training an end classifier. The experiments on several domains (classification of YouTube comments, wine quality, and weather prediction) result in considerable gains. The new approach produces significant performance improvement, leading up to +21 points in accuracy and +61 points in F1 scores compared to the state-of-the-art data programming approach.

## 1 INTRODUCTION

Supervised machine learning has proven to be very powerful and effective for solving various classification problems. However, training fully-supervised models can be costly since many applications require large amounts of labeled data. Manually annotating each data point of a large dataset may take up to weeks or even months. Furthermore, only domain experts can label the data in highly specialized scenarios such as healthcare and industrial production. Thus, the costs of data labeling might become very high.

In the past few years, a new weak supervision (WS) approach, namely data programming [Ratner et al. (2016; 2017)], has been proposed to significantly reduce the time for dataset preparation. In this approach, a domain expert writes heuristic functions named labeling functions (LFs) instead of labeling each data point. Each function annotates a subset of the dataset with an accuracy expected to be better than a random prediction. Data programming has been successfully applied to various classification tasks. However, writing LFs might not always be trivial, for instance, when data points are huge vectors of numbers or when they are not intuitively understandable. Developers can quickly code a few simple functions, but having heuristics to cover many corner cases is still a burden. Further, simple heuristics might cover only a tiny portion of the unlabeled dataset (small coverage problem).

The existing data programming framework Snorkel [Ratner et al. (2016; 2017)] implements a machine learning pipeline as follows. The LFs are applied to the unlabeled data points and the outcomes of LFs produce a labeling matrix where each data point might be annotated by multiple, even conflicting, labels. A generative model processes the labeling matrix to make single label predictions for a subset of data points, based on the agreements and disagreements on the LF outputs for a given data point $x^{(i)}$ using techniques such as majority voting (MV) or minimizing the marginalized log-likelihood estimates. Later, the label predictions are used to train a supervised model that serves as the end classifier (discriminative model). This approach has two major limitations: 1) **Coarse information** about the dataset (*only LF outputs*) fed to the generative model, 2) **Lack of generalization** due to the sparsity of labeling matrix and relying only on end classifier to generalize. The current data programming does not take the data points' data features into account during the generative process, even though they are available throughout the pipeline. It utilizes the data features of only the data

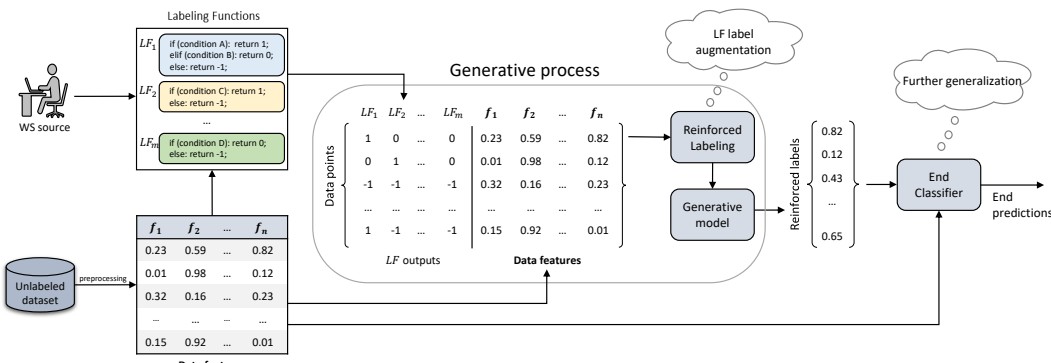

Figure 1: The label augmentation pipeline that brings data features $f_1, f_2, \ldots, f_n$ and weakly-supervised labels $LF_1, LF_2, \ldots, LF_m$ together in the generative process.

points with label predictions to train the end classifier. Various additional techniques are considered to complement the existing approach [Varma & Ré (2018); Chatterjee et al. (2020); Varma et al. (2019); Nashaat et al. (2018); Varma et al. (2016)] and improve the learning process. However, these approaches do not address this major problem in the design of existing data programming.

This article proposes a label augmentation approach for weak supervision that takes the data features and the LF outputs into account earlier in the generative process. The proposed approach utilizes the features for augmenting labels. The augmentation algorithm, namely *reinforcement labeling (RL)*, checks similarities between data features to project existing LF outcomes to non-existing labels in the matrix (i.e., to unknown cases or abstains). Moreover, it uses a heuristic that considers unknown cases "gravitate" towards the known cases with LF output labels. In such a way, RL enables *generalization early on* and creates a "reinforced" label set to train an end classifier.

Label augmentation extends the data programming to new scenarios, such as when LFs have low coverage, domain experts can implement only a limited number of LFs, or LFs outcome result in a sparse labeling matrix. Label augmentation can provide satisfactory performances in these cases, although data programming was previously non-applicable. The proposed approach can reduce the time spent by the domain experts to train a classifier as they need to implement fewer LFs. One advantage compared to the existing complementary approaches is that RL does not require any additional effort for labeling data, annotating data, or implementing additional LFs. In other words, the label augmentation enhances classification without any further development burden or assumption of available labeled datasets (e.g., the so-called "gold data"). Furthermore, one can easily combine this approach with the existing solutions.

The RL method is implemented and tested compared to Snorkel (Sn) using different fully-supervised models as end classifiers. The experimental results span classification tasks from several domains (YouTube comments, white/red wine datasets, weather prediction). The new approach outperforms the existing model in terms of accuracy and F1 scores, having closer outcomes to the fully-supervised learning, thanks to the improved coverage that enables end classifier convergence.

## 2 METHOD DESCRIPTION

### 2.1 BACKGROUND ON DATA PROGRAMMING

In data programming [Bach et al. (2019); Ratner et al. (2017)], a set of LFs annotate a portion (subset) of the original unlabeled dataset $X = \{x^{(1)}, x^{(2)}, \ldots, x^{(k)}\}$ with a total labeling coverage of $\gamma * |X|$, where $\gamma \in [0, 1]$. Given a data point $x^{(i)}$, an $LF_j$ takes $x^{(i)}$ as input and annotates the input with a label. LFs are considered weak supervisors implemented by application developers, and they can programmatically annotate many data points at once, as opposed to hand-labeling data points one by one. On the other hand, LFs may have lower accuracies than ground truth for the data points.

For a binary classification task, an LF may return two classes or abstain from making a prediction. For simplicity, let us consider $LF_j$ returning 1 or 0 as the class labels and -1 (abstain) when it refrains

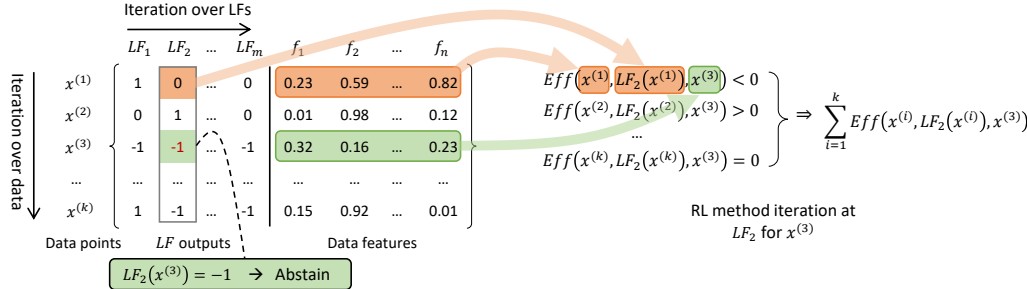

Figure 2: Illustration of the reinforced labeling method (see Alg. 1 in Supp. material A.3).

from class prediction. LFs' outputs form a labeling matrix where rows represent the data point indices $x^{(1)}, x^{(2)}, \ldots, x^{(k)}$, and columns represent the LF indices $LF_1, LF_2, \ldots, LF_m$. A generative model takes the labeling matrix as input, filters out the data points with no label (all LFs voted for abstains), and tries to predict a label for the remaining data points. An example of a generative model might be a majority voter (MV) based on LF outputs or by minimizing the negative log marginal likelihood [Ratner et al. (2017)] (likelihood over latent variables, i.e., LF outputs). Dependency structures between the LFs are learned as shown in Alg. 1 in [Bach et al. (2017)]. In both MV and marginal likelihood approaches, the generative model takes the labeling matrix as input and tries to make a label decision based on the agreements or disagreements of LFs. The generative model may fail to make a decision in certain cases, such as when equal numbers of LFs disagree on a data point. Lastly, the features of the weakly-labeled data points within $X$ and the labels from the generative model are used to supervise an end classifier (discriminative) model. The design of the data programming model is agnostic to the end classifier (discriminative) model, so various supervised machine learning models can be candidates as the end model.

## 2.2 LABEL AUGMENTATION AND REINFORCED LABELS

In the described design of data programming, the two limitations mentioned above (in Sec. 1), namely the coarse information in the labeling matrix and sparsity, lead to failure for generalizing to new and unseen data points. Therefore, these limitations may lead to reduced performances. In such scenarios, the outputs of the existing generative model may not be satisfactory to train the end classifier. Therefore, the end classifier model may not converge or generalize well enough to cover different cases. Implementing many LFs that cover different cases is costly and not very straightforward in most scenarios. Although various additional techniques focus on the weak supervision problem [Varma & Ré (2018); Chatterjee et al. (2020); Varma et al. (2019); Nashaat et al. (2018); Varma et al. (2016)], they rather extend the existing pipeline with additional features. On the other hand, label augmentation targets these major limitations by eliminating the sparsity using data features. Thus, it can lead to higher accuracy.

Fig. 1 illustrates the new pipeline for label augmentation. The new generative process brings together the outputs of $\langle LF_1, LF_2, \ldots, LF_m \rangle$, and the data features $f_1, f_2 \ldots, f_n$ of data points $x^{(i)} = \{x_1^{(i)}, x_2^{(i)}, \ldots, x_n^{(i)}\}$ in the unlabeled dataset early on. Different methods can utilize data features in the generative process for augmenting the labels in the labeling matrix. This label augmentation approach differs from the existing "data augmentation" approaches [Wang et al. (2019); Tran et al. (2017); Cubuk et al. (2019)] that create new (synthetic) data points, as the new goal is to project LF outcomes to the existing data points as opposed to creating new data points.

The outcome of the new generative process can be more representative of the data and weak supervisors than the outputs of the previously existing generative process due to additional coverage and accuracy gains without any additional LF implementation or data annotation. For instance, abstain values (-1s) from the LFs' outputs representing the unknown cases (left side of the matrix in Fig. 1) can be predicted by the new generative process (as outlined in Sec. 2.3). The abstain values in the labeling matrix can be augmented with classification prediction values.

Data programming applies statistics only to data points that are already covered by LFs, resulting in a single predicted label for a subset of the dataset. Similarly, by using likelihood estimation over the

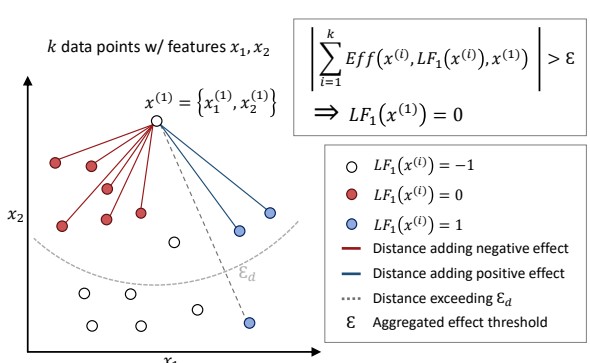 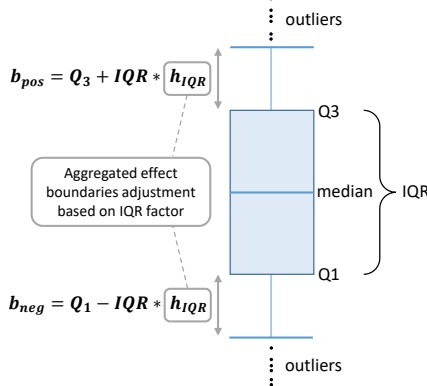

Figure 3: Illustration of the gravitation-based RL method

Figure 4: The IQR factor adjustment to dynamically select $\varepsilon$ for gravitation.

augmented matrix, the generative process predicts "reinforced labels". The details of the generative model implementation can be seen in [Bach et al. (2017)]. After the generative process, the reinforced labels are used to train the end classifier. The end classifier can still be the same supervised machine learning model. As the augmented matrix has more density compared to the labeling matrix, it may lead to a larger training set. As a simple example, a disagreement between two LFs for a data point can be eliminated by a new prediction for the abstain case of a third LF. As a result, more data points can be used to train the end model. The benefits of using the end classifier include further generalization for the data points that are still not labeled. Furthermore, the reinforced labels can fine-tune pretrained machine learning models.

## 2.3 IMPLEMENTING REINFORCED LABELING

Let us describe the RL method for label augmentation in the generative process. The intuition behind this method is that a data point $x^{(j)}$, that a LF does not label, might have very similar features compared to another data point, such as a data point $x^{(i)}$ labeled by the same LF. $x^{(j)}$ might not be labeled due to the conditions or boundaries in the LF or missing a subset of the data features in its heuristic implementation. Reinforcing the labels means predicting a label for the previously unlabeled data points by certain LFs and therefore creating a denser version of the labeling matrix during the generative process. Furthermore, RL could lead to a higher coverage $\gamma' * |X| \geq \gamma * |X|$ for training the end classifier.

Fig. 2 illustrates the RL method. The method iterates over $\langle LF_1, LF_2, \ldots, LF_m \rangle$ and for each $LF_l$ on the left part of the matrix, it finds the data points $x^{(j)}$s that $LF_l$ abstains from (-1s on the matrix). The abstain points are compared with the points that are labeled by $LF_l$ for their distances based on data features. The similarity between a point $x^{(i)}$ labeled by $LF_l$ point and the unlabeled $x^{(j)}$ is represented by the "effect function" $Eff(x^{(i)}, LF_l(x^{(i)}), x^{(j)})$. As inputs, the effect function has the data features of the labeled $x^{(i)} = \{x_1^{(i)}, x_2^{(i)}, \ldots, x_n^{(i)}\}$, output label $LF_l(x^{(i)})$ and the features of the unlabeled $x^{(j)} = \{x_1^{(j)}, x_2^{(j)}, \ldots, x_n^{(j)}\}$. Based on the label, the effect may take a positive value (if $LF_l(x^{(i)}) = 1$) or a negative value (if $LF_l(x^{(i)}) = 0$). For any other data point that the $LF_l$ also abstains, it takes the value zero. These values can be aggregated as $\sum_{i=1}^{k} Eff(x^{(i)}, LF_l(x^{(i)}), x^{(j)})$ for any $LF_l$ and where $k = |X|$.

We propose a heuristic algorithm to implement RL based on gravitation. Fig. 3 illustrates the heuristic method considering a simple case of having only two data features $x_1, x_2$. In the gravitation-based RL method, for the abstain point $x^{(j)}$ that is not labeled by $LF_l$, each other point that $LF_l$ labels (shown as colored particles in Fig. 3) is considered as a particle that attracts $x^{(j)}$ towards positive ($LF_l(x_{(i)}) = 1$) or negative ($LF_l(x_{(i)}) = 0$) aggregated effect. Fig. 3 shows positive or negative attractions by different-colored lines between the data points. The attraction is inversely proportional to the pairwise distance of every labeled point $x^{(i)}$ to $x^{(j)}$. For optimizing the calculations, we

consider a maximum distance threshold $\varepsilon_d$ above which attraction does not take place. Hence, the effect function is defined as follows:

$$Eff\left(x^{(i)}, LF_l(x^{(i)}), x^{(j)}\right) = \begin{cases} \frac{\beta}{Distance\left(x^{(i)}, x^{(j)}\right)^\alpha} & \text{if } Distance(x^{(i)}, x^{(j)}) < \varepsilon_d \\ 0 & \text{otherwise,} \end{cases} \tag{1}$$

where $\alpha$ and $\beta$ : are constants for optional adjustment of the effect and distance relationship. As a distance function $Distance(x^{(i)}, x^{(j)})$ RL can use different metrics based on the data types (e.g., sensor readings, text, image) such as Euclidean, Levenshtein or Mahalanobis distances. The effects of all labeled points are aggregated based on their positive or negative attractions. The decision of updating an abstain value depends upon a parameter called the *aggregated effect threshold* $\varepsilon$. $x^{(j)}$ is labeled with that class for the given $LF_l$ if the following condition holds:

$$\left| \sum_{i=1}^{|X|} Eff\left(x^{(i)}, LF_l(x^{(i)}), x^{(j)}\right) \right| > \varepsilon. \tag{2}$$

$\varepsilon$ adjusts the degree of reinforcement and the density of the resulting augmented matrix. In the evaluation section, we consider various empirical parameters and a heuristic for automatic $\varepsilon$ configuration.

## 3 EXPERIMENTAL EVALUATION

### 3.1 BENCHMARK FRAMEWORK

The experimental study includes four metrics: **Classification accuracy** (# correct/# tested), **precision**, **recall**, and **F1 score** (F1). Particularly, higher accuracy and F1 score are important to understand the performance for the given classification task for non-biased and biased tasks, respectively. Other than these performance metrics, the experimental evaluation includes insights on the labeling itself (labeling metrics) such as: **Number of LFs:** The number of LFs used as weak supervisors; **Labeled samples:** The number of samples labeled for training the end models; **LF coverage:** The ratio of the number of data points labeled by LFs to the number of samples; **LF overlap:** The ratio of LFs' overlapping outputs with each other to the number of samples; **LF conflicts:** The ratio of LFs' conflicting outputs with other each other to the number of samples. The last three metrics represent the mean values, averaged based on the number of LFs.

We evaluate the RL approach with four datasets from different domains: YouTube comments [Alberto et al. (2015)], red and white wine quality [Cortez et al. (2009)], and Australia Weather (rain) datasets[1]. YouTube comments dataset consists of texts that may be legit user comments or spam. This text-based dataset is used for benchmarking various data programming approaches [Chen et al. (2020); Evensen et al. (2020); Ren et al. (2020); Karamanolakis et al. (2021); Sedova et al. (2021); Awasthi et al. (2020)] and also as Snorkel's tutorial for LFs. The YouTube dataset is largely unlabeled except for a small testing dataset. Only the text comment is used as a feature for the YouTube dataset, whereas we removed the others (such as user ID and timestamp). On the other hand, the end models get a sparse matrix of token counts computed with Scikit-learn CountVectorizer[2]. To classify the wine quality, there exist 12 real number features (e.g., acidity, residual sugar) for the two wine datasets. For training and testing, a wine is considered good (labeled with 1) when the wine quality feature is more than 5 out of 10; otherwise considered a bad wine (0). The Australia Weather dataset is widely used for the next-day rain predictions. It also consists of 62 features based on daily weather observations spanning ten years period. The wine and Australia Weather datasets are fully labeled with ground truth. For all datasets Euclidean distance metric computes distances between data point pairs. For the YouTube dataset, Euclidean distance is applied to the one-hot encoding of the tokenized texts[3].

Snorkel (Sn) and fully-supervised learning (Sup) are the two main approaches for comparison. In addition to those, the majority voting (MV) approach is tested as the simple generative model. For

---

[1]https://www.kaggle.com/jsphyg/weather-dataset-rattle-package
[2]https://scikit-learn.org/stable/modules/generated/sklearn.feature_extraction.text.CountVectorizer.html
[3]https://www.nltk.org/api/nltk.tokenize.html

| | Reinforced labeling | | | | | Snorkel | | | | Supervised learning | | | |
|---|---|---|---|---|---|---|---|---|---|---|---|---|---|
| **Dataset** | **Acc** | **Prec** | **Rec** | **F1** | **F1-Gain** | **Acc** | **Prec** | **Rec** | **F1** | **Acc** | **Prec** | **Rec** | **F1** |
| YouTube | 0.75 | 0.98 | 0.47 | 0.64 | **+61** | 0.54 | 1.00 | 0.02 | 0.03 | 0.91 | 0.96 | 0.84 | 0.90 |
| Red Wine | 0.71 | 0.80 | 0.66 | 0.72 | **+7** | 0.61 | 0.66 | 0.77 | 0.65 | 0.75 | 0.81 | 0.74 | 0.76 |
| White Wine | 0.63 | 0.64 | 0.98 | 0.78 | **+34** | 0.50 | 0.82 | 0.32 | 0.44 | 0.54 | 0.71 | 0.48 | 0.57 |
| Weather | 0.59 | 0.29 | 0.78 | 0.42 | **+34** | 0.54 | 0.06 | 0.10 | 0.08 | 0.90 | 0.86 | 0.59 | 0.70 |

Table 1: RL, Snorkel, and (fully-)supervised learning results: Accuracy, precision, recall, and F1. F1-Gain shows the F1 score advantage of RL compared to Snorkel.

| | Reinforced labeling | | | | | | Snorkel | | | | All | |
|---|---|---|---|---|---|---|---|---|---|---|---|---|
| **Dataset** | **RL labels** | **LF cov.** | **LF ov.** | **LF con.** | $\varepsilon$ | $\varepsilon_d$ | **Sn labels** | **LF cov.** | **LF ov.** | **LF con.** | **End model** | **LFs** |
| YouTube | 1273 | 0.22 | 0.11 | 0.04 | 75 | 5.0 | 916 | 0.16 | 0.08 | 0.03 | svm | 5 |
| Red Wine | 375 | 0.12 | 0.02 | 0.02 | 125 | 0.5 | 247 | 0.08 | 0.02 | 0.01 | RF | 3 |
| White Wine | 3269 | 0.39 | 0.15 | 0.07 | 350 | 0.5 | 1995 | 0.21 | 0.04 | 0.01 | NaivB | 3 |
| Weather | 3415 | 0.58 | 0.56 | 0.49 | 200 | 5.0 | 2384 | 0.19 | 0.13 | 0.10 | RF | 6 |

Table 2: Labeling metrics for the approaches and dataset statistics.

each data point, MV labels the point based on the majority of the LF outputs (i.e., 0 or 1) from the LFs that do not abstain. We also experiment with MV combined with RL (MVRL). For all results, the existing data programming approaches Sn and MV use the same set of LFs as RL. The LFs are listed in Supp. material A.5. We implement Sn and MV as well as the RL approach using the Snorkel library (version 0.9.5) with the existing features in the tutorial. The framework uses absolute latent labels for training end model (0,1), RL adapts the same scheme for the generative model and training end classifiers. Supervised learning leverages ground truth data (30% of the dataset) for the training. For the experimental scenario, supervised learning is considered the optimal result of machine learning, whereas the other two models are based on only programmatic labels. For supervised learning, the small testing dataset of YouTube comments is used for both training and testing due to lacking ground truth in the training set.

For the end classifier, different machine learning models are used for testing purposes. The models include two logistic regression models: the first one, namely "logit", is the model that Snorkel uses by default (inverse of regularization strength $C := 1000$ and liblinear solver); the second one, "LogR", uses lbfgs solver for optimization. In addition, we test the random forest (RF), naive Bayes (NaivB), decision tree (DT), k-nearest neighbor (knn), support vector machine (svm), and multi-layer perceptron (mlp) end models. In each experiment, all approaches use the same end classifier. Each experiment consists of 5 runs, and the results are averaged. We do not observe any notable difference between experiment runs. Our assumption is that the tested end models learning behavior is rather deterministic as they use the same the data points for their trainings. There are no additional hyperparameters used other than the stated in this section.

## 3.2 EXPERIMENTAL RESULTS

**Benefits of RL** The first set of results shows the advantage of using RL compared to the existing data programming approach Sn in terms of accuracy and F1 score gains. Table 1 includes results from the four datasets in terms of the four performance metrics. In all datasets, we observe substantial gains in accuracy and F1 scores compared to the benchmark Sn approach. Moreover, RL performance is closer to fully-supervised learning, although it does not use any ground truth labels, even when LFs are relatively few. For the YouTube dataset, we experience that although the Snorkel and RL approaches have the same set of LFs, RL provides up to 64% F1, whereas Sn provides less than 3% F1. In RL, the end model can converge thanks to the additional coverage by the reinforced labels. Table 2 reports the number of computed labels after the application of the generative model with reinforcement (RL) and without it (Snorkel). The gains for accuracy and F1 require no additional human effort or involvement. Only two parameters $\varepsilon$ and $\varepsilon_d$ configure the RL (see Table 2 for Table 1 experiments settings). Later in this section, we present how these parameters could be configured automatically for any given scenario based on labeling metrics (i.e., LF coverage, LF overlaps, and LF conflicts). Furthermore, our approach consistently outperforms Sn for training any of the tested end models. An extended table in Supp. material A.1 reports the experimental results varying end

models for each dataset. Lastly, we observe that $\varepsilon_d$ affects the performance as it adjusts the trade-off between augmenting similar labels and the bias (noise) of the augmentation.

**Auto-adjustment of the RL method** We observe that $\varepsilon$ is dependent on the dataset and the set of LFs. A simple heuristic method to choose $\varepsilon$ is as follows: First calculate the distribution of aggregated effects of unlabeled data point for each LF (through a boxplot as in Fig. 4), then set $\varepsilon$ to have boundaries $b_{neg}$ and $b_{pos}$ far enough from the quartiles $Q_1$ and $Q_3$. When using 2 such boundaries are symmetric to aggregate effect equal to 0 and they are chosen by the data scientist as we did for the results shown in Tables 1. Another way to calculate such boundaries is to have them symmetrical to the aggregated effect distribution. We can achieve this with the formula $b_{neg} = Q_1 - IQR * h_{IQR}$ (and similarly for $b_{pos}$) where $InterQuartileRange(IQR) = Q_3 - Q_1$ and $h_{IQR}$ is a parameter, namely the IQR factor. When $h_{IQR} = 1.5$, the gravitation method labels only outliers of the aggregated effect distribution. At this point, there is no need to set the $\varepsilon$ parameter and the boundaries automatically adapts to the aggregated effects that depends on the features range, distance metrics and the sparsity of the initial labeling matrix. However, varying $h_{IQR}$ (as tested for the range $[0, 2]$) affects the performances. Thus, the problem of finding an optimal $h_{IQR}$ still persists.

Fig. 5a-top row shows the comparison with Snorkel generative model, while Fig. 5a-middle row shows the comparison with MV. As in Fig. 5a-top, $h_{IQR} \leftarrow 0.5$ cause a substantial F1 gain for fewer LFs (e.g., 5 to 8 LFs), whereas it may cause detrimental bias for the case of a higher number of LFs as the sum of LFs' coverage increases. In this case, the reinforcement may not need to be applied as extensively as when the sum of LFs' coverage is low. In the latter scenario, using $h_{IQR} \leftarrow 1.0$ would be the most conservative approach (no reinforcement) that makes sure to add no additional noise. Similar behavior is observed with majority voter as label aggregation algorithm (Fig. 5a-middle row).

Fig. 5a-bottom row shows the effect of RL with different $h_{IQR}$ in terms of labeled samples, LFs' coverage, mean overlaps, and mean conflicts. The smaller is IQR factor, the higher the values for those metrics are because the boundaries are closer to the IQR. One can infer that for a higher number of LFs, a smaller IQR factor results in excessive noise confusing the generative model (Sn or MV) and degrading the end model. Thus, we define a simple heuristic to automatically configure the gravitation method (shown in Alg. 2 in Supp. material A.3) through calculating $h_{IQR}$ by linking it to the LF statistics for the given dataset. The below formula uses these statistics and an empirical constant $\xi$.

$$h_{IQR} = \xi * \sum_{LF_l} coverage * \sum_{LF_l} overlaps * \sum_{LF_l} conflicts \qquad (3)$$

We set $\xi = 0.35$ through empirical experiments. By this approach, the higher density the labeling matrix has (e.g., more LFs, higher LF coverage), the fewer the abstain labels updated by RL are.

We test the auto-adjustment approach and show the results in Fig. 5b for both Sn (top row) and MV (middle row). We have also set the second parameter $\varepsilon_d$ virtually to infinite, relying on no hyper-parameter. When fewer LFs exist (e.g., 5, 6, or 7), RL provides significant performance gains, especially for F1 scores. On the other hand, when more LFs exist, RL adjusts itself by becoming more conservative on the reinforcement. Therefore, auto-adjustment preserves the good performances when having sufficient LFs, leading to a larger area under the curve for both Sn and MV. Our proposed RL approach, together with this heuristic, does not require any additional parameter setting or human effort by automatically adjusting its IQR factor. However, additional studies are due for more enhanced methods of auto-adjustment.

Fig. 5b shows the effect of the automatic IQR adaptation in terms of labeled samples and LF metrics. As expected, when fewer LFs exist, RL provides gains for the number of labeled data points and mean LF coverage, overlaps, and conflicts. Furthermore, it adapts itself and gradually provides lower numbers of additional labels for higher numbers of LFs, thus reducing the noise.

**Varying distance metrics** Our gravitation method highly rely on similarities between data points. In the experiments described above, we always use the Euclidean distance for the real number features. Therefore, we investigate how different distance metrics affect the performances. Table 3 shows the results on the white wine dataset with RF as the end classifier and the previous heuristics to calculate the aggregated effect boundaries with $h_{IQR} \leftarrow 1.4$. RL consistently provides improved performance even when the distance metric changes.

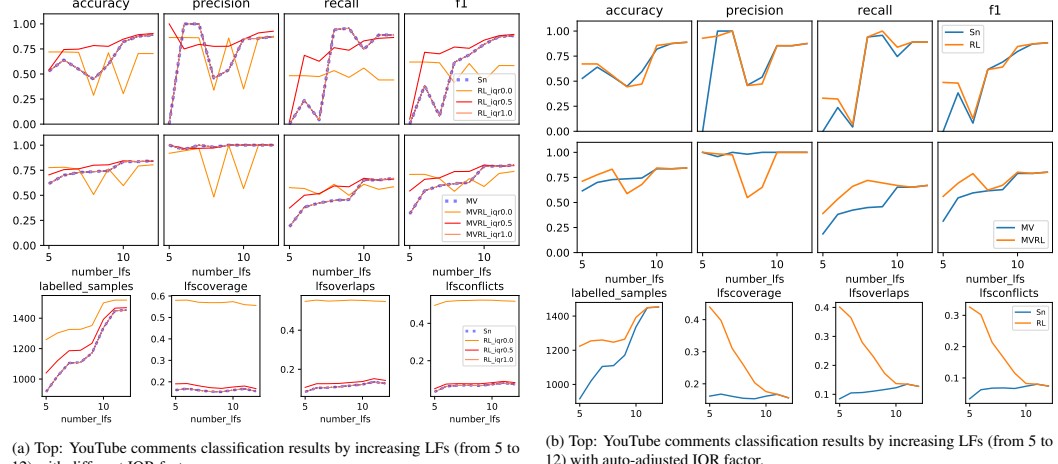

(a) Top: YouTube comments classification results by increasing LFs (from 5 to 12) with different IQR factors.
Bottom: RL effects on the total number of labeled samples, and mean LF coverage, overlaps, and conflicts.

(b) Top: YouTube comments classification results by increasing LFs (from 5 to 12) with auto-adjusted IQR factor.
Bottom: effects of RL with auto-adjusted IQR factor on the total number of labeled samples and mean LFs coverage, overlaps, and conflicts.

Figure 5: (a) Effects of the $h_{IQR}$ by the number of LFs. (b) RL with the auto-adjusted $h_{IQR}$.

| | | Reinforced LFs | | | | | Snorkel | | |
| Distance metric | Acc | Prec | Rec | F1 | F1-Gain | Acc | Prec | Rec | F1 |
| --- | --- | --- | --- | --- | --- | --- | --- | --- | --- |
| Chebyshev | 0.63 | 0.65 | 0.94 | 0.77 | **+20** | 0.53 | 0.71 | 0.48 | 0.57 |
| Cosine | 0.66 | 0.67 | 0.96 | 0.79 | **+25** | 0.51 | 0.70 | 0.44 | 0.54 |
| Euclidean | 0.66 | 0.68 | 0.92 | 0.78 | **+23** | 0.51 | 0.70 | 0.46 | 0.55 |
| Hamming | 0.62 | 0.66 | 0.88 | 0.75 | **+16** | 0.52 | 0.68 | 0.52 | 0.59 |
| Jaccard | 0.60 | 0.67 | 0.79 | 0.71 | **+14** | 0.51 | 0.69 | 0.49 | 0.57 |
| Mahalanobis | 0.64 | 0.66 | 0.94 | 0.78 | **+24** | 0.51 | 0.70 | 0.44 | 0.54 |
| Minkowski | 0.57 | 0.68 | 0.67 | 0.66 | **+9** | 0.52 | 0.69 | 0.49 | 0.57 |

Table 3: Testing RL with various distance metrics for WhiteWine: RF as end model, $h_{IQR} = 1.4$.

**Limitations** As label augmentation relies on the data features, using a poor embedding for distance computation might lead to detrimental noise in the augmented labeling matrix. In addition, a more advanced augmentation adjustment would avoid performance degradation due to the tradeoff between the coverage and noise.

## 4 RELATED WORK

**Data programming** [Ratner et al. (2016)] enables programmatically labeling data and training using WS sources. In particular, the Snorkel framework [Ratner et al. (2017); Bach et al. (2019)] provides an interface for data programming where users can write LFs and apply them to generate a training dataset for their end models. The generation of the training dataset relies on the generative model, and several studies focus on this aspect [Ratner et al. (2017); Bach et al. (2017); Varma et al. (2017; 2019)].

Various recent works focus on extending the existing data programming approach. The extensions include multi-task classification [Ratner et al. (2019)], using small labeled gold data for augmenting WS [Varma & Ré (2018)], learning tasks (or sub-tasks) in slices of dataset [Chen et al. (2019)], user guidance for LF precision [Chatterjee et al. (2020)], making the training process faster [Fu et al. (2020)], generative adversarial data programming [Pal & Balasubramanian (2020; 2018)], user supervision for LF error estimates [Arachie & Huang (2019)], learning LF dependency structures [Varma et al. (2019)], user annotation of LFs [Boecking et al. (2021)], language description of LFs [Hancock et al. (2018)], active learning Nashaat et al. (2018), and so on. Varma et al. (2016) aims to learn common features by using a "difference model" and feeding these features back to generative model. Mallinar et al. (2019) takes advantage of the natural language processing query engine to expand gold labels and generate a label matrix as input for the generative model. Zhou et al. (2020) adopts a soft LFs matcher approach based on the distances between LFs' conditions and data points.

Chen et al. (2020) uses pre-trained machine learning models to estimate distances for natural language processing. The last two studies focus on the semantic similarity of texts to improve the labeling.

Although the studies mentioned above may improve the existing generative model (e.g., through additional human supervision), they do not focus on the problem of LF abstraction with coarse information. Solving this problem would improve the validity of data programming in various scenarios especially since the human supervision is limited in its nature. In this paper, we identify this problem and propose the reinforced labeling that takes the data features into account early on in the generative process. Using this approach, one can leverage the data features and augment the matrix for further generalization and producing satisfactory performance *without additional human supervision*.

**Weak supervision** approaches outside the context of data programming consider learning from a set of rules, experts, or workers as in crowdsourcing. Platanios et al. (2017) infer accuracy from multiple classifier outputs based on the confidences. Safranchik et al. (2020) study the usage of Hidden Markov Models for tagging data sequences. Dehghani et al. (2018) train deep NNs using weakly-labeled data. Their approach is semi-supervised, where a teacher network based on rules adjusts the predictions of a student network trained iteratively by given data samples. In another study, Dehghani et al. (2017) propose WS to train neural ranking models in natural language processing and computer vision tasks.

Takeoka et al. (2020) consider leveraging unsure responses of human annotators as WS sources to improve the traditional supervised approach. Kulkarni et al. (2018) study labeling based on consensus and interactive learning based on active labeling for multi-label image classification tasks. Khetan et al. (2018) propose an expectation-maximization algorithm for learning workers' quality, where each worker represents a WS source for image classification tasks. Qian et al. (2020) propose WS with active learning for learning structured representations of entity names. Guan et al. (2018) learn individual's weights for predicting a weighted sum of noisy labelers (experts). Das et al. (2020) propose a domain-agnostic approach to replace the needs of LFs that apply affinity functions to relate samples with each other. This approach uses a small gold dataset with probabilistic distributions to infer probabilistic labels.

**Similar approaches other than WS** include domain-specific machine learning applications such as ontology matching. Doan et al. (2004) use a relaxation method to label a node into a graph dataset by analyzing features of the node's neighborhood in the graph. The relaxation process is based on constraint and knowledge that leads to the final labeling. In a similar approach, Li & Srikumar (2019) describe a methodology framework to augment labels guided by external knowledge. In both approaches, label augmentation is in the final phase. These approaches do not involve any generative process. Lastly, literature related to semi-supervised learning (e.g., [Zhu et al. (2003)]) or other hybrid approaches (e.g., [Awasthi et al. (2020)]) consider using a mix of clean and noisy labels, whereas this paper focuses on using *only* the labels from LFs and improve the validity of the existing approach.

## 5 CONCLUSION

This paper proposes a novel method for label augmentation in weak supervision. In the new machine learning pipeline, the proposed RL method in the generative process leverages existing LF outputs and data features to augment the weakly-supervised labels. The experimental evaluation shows the benefits of RL for four classification tasks compared to the existing data programming approach in terms of substantial accuracy and F1 gains. Furthermore, the new method enables the convergence of the end classifier even when there exist few LFs. We consider applying RL for matching problems (e.g., entity matching) and active learning as future work. We consider RL as an initial approach for the identified limitation of the generative process, whereas the pipeline opens up the possibility for more advanced (e.g., machine learning) models to leverage data features during the generative process.

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

## 6 REPRODUCIBILITY STATEMENT

The results shown in the paper and the appendix can be reproduced as described below.

All of the four datasets are openly available and popularly used by many machine learning researchers. The open-source Snorkel framework tutorial [Snorkel (2021)] (version 0.9.5) can be run as it is, as the existing data programming approach. In the experiments the default features in these tutorial is used without any change, whereas we implemented new LFs for the new datasets and include all of these new LFs in Appendix A.5. The algorithm implemented for reinforcement of labels and IQR calculation are included in detail in Appendix 1 as Alg. 1 and Alg. 2. We plan to make the codes for these two algorithms open-source after our internal approval process.

A person skilled in programming (e.g., Python) can implement the two algorithms in a short time given the pseudocodes. The only two parameters ($\varepsilon$ and $\varepsilon_d$ in Table 1) used in the four experiments are listed in Table 2 and four distance metrics, instead of these two parameters, only one parameter value $h_{IQR} := 1.4$ (for Alg. 2) is used as shown in Table 3.

# A APPENDIX

## A.1 REINFORCED LABELING RESULTS ON DIFFERENT DATASETS VARYING END MODELS

Table 4 extends Table 1 by reporting the results of 8 end models and 4 datasets. The hyperparameter configurations of $\varepsilon$ and $\varepsilon_d$ are listed in Table 2. For all experiments and end models RL outperforms Snorkel and maintains closer performance to the fully-supervised machine learning.

| Dataset | Reinforced LFs | | | | | Snorkel | | | | Supervised learning | | | | End model |
|---|---|---|---|---|---|---|---|---|---|---|---|---|---|---|
| | Acc | Prec | Rec | F1 | F1-Gain | Acc | Prec | Rec | F1 | Acc | Prec | Rec | F1 | |
| YouTube | 0.75 | 0.98 | 0.47 | 0.64 | **+61** | 0.54 | 1.00 | 0.02 | 0.03 | 0.91 | 0.96 | 0.84 | 0.90 | svm |
| YouTube | 0.72 | 1.00 | 0.40 | 0.57 | **+55** | 0.53 | 1.00 | 0.01 | 0.02 | 0.81 | 1.00 | 0.61 | 0.76 | LogR |
| YouTube | 0.68 | 0.98 | 0.34 | 0.50 | **+50** | 0.53 | 0.00 | 0.00 | 0.00 | 0.92 | 0.92 | 0.90 | 0.91 | DT |
| YouTube | 0.72 | 1.00 | 0.41 | 0.58 | **+56** | 0.53 | 1.00 | 0.01 | 0.02 | 0.91 | 1.00 | 0.82 | 0.90 | logit |
| YouTube | 0.67 | 0.93 | 0.33 | 0.49 | **+44** | 0.53 | 0.50 | 0.03 | 0.05 | 0.76 | 1.00 | 0.53 | 0.69 | knn |
| YouTube | 0.69 | 1.00 | 0.34 | 0.51 | **+51** | 0.53 | 0.00 | 0.00 | 0.00 | 0.86 | 1.00 | 0.70 | 0.82 | RF |
| Red Wine | 0.71 | 0.82 | 0.62 | 0.70 | **+1** | 0.70 | 0.80 | 0.64 | 0.69 | 0.74 | 0.82 | 0.70 | 0.74 | svm |
| Red Wine | 0.68 | 0.85 | 0.53 | 0.64 | **+3** | 0.68 | 0.85 | 0.55 | 0.61 | 0.73 | 0.80 | 0.71 | 0.74 | LogR |
| Red Wine | 0.69 | 0.78 | 0.63 | 0.69 | **+4** | 0.60 | 0.65 | 0.78 | 0.65 | 0.65 | 0.70 | 0.65 | 0.67 | DT |
| Red Wine | 0.71 | 0.79 | 0.68 | 0.72 | **+2** | 0.68 | 0.73 | 0.74 | 0.70 | 0.73 | 0.82 | 0.69 | 0.73 | logit |
| Red Wine | 0.71 | 0.82 | 0.61 | 0.70 | **+5** | 0.68 | 0.80 | 0.59 | 0.65 | 0.75 | 0.80 | 0.75 | 0.77 | knn |
| Red Wine | 0.69 | 0.69 | 0.81 | 0.74 | **0** | 0.66 | 0.63 | 0.89 | 0.74 | 0.71 | 0.77 | 0.67 | 0.71 | NaivB |
| Red Wine | 0.71 | 0.80 | 0.66 | 0.72 | **+7** | 0.61 | 0.66 | 0.77 | 0.65 | 0.75 | 0.81 | 0.74 | 0.76 | RF |
| Red Wine | 0.70 | 0.83 | 0.60 | 0.69 | **+1** | 0.70 | 0.83 | 0.61 | 0.68 | 0.74 | 0.82 | 0.71 | 0.74 | mlp |
| White Wine | 0.65 | 0.65 | 1.00 | 0.79 | **+23** | 0.51 | 0.67 | 0.48 | 0.56 | 0.72 | 0.73 | 0.91 | 0.81 | svm |
| White Wine | 0.65 | 0.65 | 1.00 | 0.79 | **+22** | 0.52 | 0.68 | 0.51 | 0.57 | 0.72 | 0.74 | 0.90 | 0.81 | LogR |
| White Wine | 0.63 | 0.64 | 0.97 | 0.77 | **+26** | 0.50 | 0.69 | 0.41 | 0.51 | 0.56 | 0.70 | 0.57 | 0.63 | DT |
| White Wine | 0.65 | 0.66 | 0.99 | 0.79 | **+23** | 0.50 | 0.67 | 0.49 | 0.56 | 0.67 | 0.86 | 0.62 | 0.69 | logit |
| White Wine | 0.64 | 0.65 | 0.99 | 0.78 | **+8** | 0.59 | 0.67 | 0.73 | 0.70 | 0.65 | 0.75 | 0.70 | 0.72 | knn |
| White Wine | 0.63 | 0.64 | 0.98 | 0.78 | **+34** | 0.50 | 0.82 | 0.32 | 0.44 | 0.54 | 0.71 | 0.48 | 0.57 | NaivB |
| White Wine | 0.64 | 0.65 | 0.99 | 0.78 | **+27** | 0.50 | 0.69 | 0.41 | 0.51 | 0.71 | 0.81 | 0.73 | 0.76 | RF |
| White Wine | 0.66 | 0.66 | 1.00 | 0.79 | **+18** | 0.54 | 0.68 | 0.56 | 0.61 | 0.71 | 0.82 | 0.71 | 0.75 | mlp |
| Australia Rain | 0.58 | 0.28 | 0.77 | 0.41 | **+31** | 0.46 | 0.07 | 0.15 | 0.10 | 0.86 | 0.75 | 0.40 | 0.52 | svm |
| Australia Rain | 0.58 | 0.28 | 0.76 | 0.41 | **+32** | 0.49 | 0.07 | 0.13 | 0.09 | 0.86 | 0.73 | 0.40 | 0.52 | LogR |
| Australia Rain | 0.55 | 0.28 | 0.83 | 0.42 | **+30** | 0.37 | 0.08 | 0.21 | 0.12 | 0.85 | 0.61 | 0.69 | 0.65 | DT |
| Australia Rain | 0.59 | 0.29 | 0.76 | 0.42 | **+32** | 0.44 | 0.07 | 0.16 | 0.10 | 0.87 | 0.74 | 0.53 | 0.61 | logit |
| Australia Rain | 0.49 | 0.25 | 0.81 | 0.38 | **+29** | 0.53 | 0.07 | 0.12 | 0.09 | 0.82 | 0.54 | 0.55 | 0.55 | knn |
| Australia Rain | 0.54 | 0.26 | 0.75 | 0.39 | **+27** | 0.50 | 0.09 | 0.18 | 0.12 | 0.69 | 0.34 | 0.65 | 0.45 | NaivB |
| Australia Rain | 0.59 | 0.29 | 0.78 | 0.42 | **+34** | 0.54 | 0.06 | 0.10 | 0.08 | 0.90 | 0.86 | 0.59 | 0.70 | RF |
| Australia Rain | 0.57 | 0.27 | 0.82 | 0.41 | **+33** | 0.48 | 0.06 | 0.12 | 0.08 | 0.87 | 0.66 | 0.55 | 0.60 | mlp |

Table 4: RL, Snorkel, and (fully-)supervised model results: Accuracy, recall, precision and F1 scores. F1-Gain shows the F1 score advantage of RL compared to Snorkel.

## A.2 SYMBOLS AND NOTATIONS IN THE PAPER

Table 5 lists and describes the frequently used symbols throughout the paper. Some listed parameters are not normalized (e.g., aggregated effects) or adjusted for simplicity, whereas they can be easily normalized to the range $[0, 1]$, and so $\varepsilon \in [0, 1]$ based on the observed values, without causing any change in the outcomes.

| Symbol | Description |
|---|---|
| $x^{(i)} = \{x_1^{(i)}, x_2^{(i)}, \ldots, x_n^{(i)}\}$ | Data point composed of $n$ features |
| $f_n$ | Data feature |
| $X = \{x^{(1)}, x^{(2)}, \ldots, x^{(k)}\}$ | Dataset |
| $\gamma \in [0, 1]$ | Dataset coverage |
| $k = |X|$ | Number of data points |
| $LF_l$ | Labeling function with index $l$ |
| $LF_l(x^{(i)}) \in \{0, 1, -1\}$ | Output of $LF_j$ on data point $x^{(i)}$. Possible outcomes are classes 0 or 1, or abstain -1 |
| $\langle LF_l \rangle = \{LF_l(x^{(1)}), ..., LF_l(x^{(k)})\}$ | Output of labeling function $LF_l$ applied on the whole dataset |
| $Eff(x^{(i)}, LF_l(x^{(i)}), x^{(j)})$ | Effect function of data points $x^{(i)}$ and $x^{(j)}$, and output of $LF_l$ on data point $x^{(i)}$ |
| $\sum_{i=1}^{k} Eff(x^{(i)}, LF_l(x^{(i)}), x^{(j)})$ | Aggregated effect on point $x^{(j)}$ for $LF_l$ |
| $Distance(x^{(i)}, x^{(j)})$ | Distance (e.g., Euclidean) between two data points |
| $\varepsilon_d$ | Cut off distance for an effect to be considered |
| $\alpha = 1, \beta = 1, \xi = 0.35$ | Constants |
| $b_{neg}, b_{pos}$ | Thresholds on the aggregated effected to augment a label as negative or positive |
| $\varepsilon = b_{pos} = -b_{neg}$ | Symmetric aggregated effect threshold |
| $Q_1, Q_2, Q_3$ | Quartiles. 25th, 50th and 75th percentile respectively. |
| $IQR = Q_3 - Q_1$ | InterQuartile Range |
| $h_{IQR}$ | IQR factor. $h_{IQR} = 1.5$ is used to calculate the outliers range |
| $coverage_l, overlaps_l, conflicts_l$ | Statistics of the $LF_l$ also in relations with all the other $LF_j$ |

Table 5: Frequently used symbols

## A.3 Reinforced labeling pseudocode for label augmentation

---

**Algorithm 1:** Reinforced labeling algorithm

---

**Input:** LFs $\langle LF_1, LF_2, \ldots LF_m \rangle$ and unlabeled data points $X = \{x^{(1)}, x^{(2)}, \ldots, x^{(k)}\}$, where
$x^{(i)}$ has features $\langle x_1^{(i)}, x_2^{(i)}, \ldots x_n^{(i)} \rangle$. Gravity parameters $\alpha, \beta$. Distance threshold $\varepsilon_d$.
IQR adjusting parameter $\xi$
**Output:** Label for a subset $\gamma' * |X|$ of the data points (augmented labels)
$Y = \{y^{(1)}, y^{(i)}, \ldots, y^{(k)}\}$, where $y^{(i)} \in [0, 1] \cup \{-1\}$

1  $Distances \leftarrow \emptyset$
2  $Effects \leftarrow \emptyset$
3  $Labels \leftarrow \emptyset$
4  **for** $i = 1, 2, \ldots, |X|$ **do**
5      **for** $l = 1, 2, \ldots, m$ **do**
6          $Labels[i][l] \leftarrow LF_l(x^{(i)})$
7          $Effects[i][l] \leftarrow 0$
8      **end**
9  **end**
10 $LFs\_stats \leftarrow CalculateLFsStats(X, Labels)$
11 **for** $l = 1, 2, \ldots, m$ **do**
12     **for** $i = 1, 2, \ldots, |X|$ **do**
13         **if** $Labels[i][l] == -1$ **then**
14             **for** $j = 1, 2, \ldots |X|$ **do**
15                 **if** $i \neq j$ & $Labels[j][l] \neq -1$ **then**
16                     $Distance \leftarrow \infty$
17                     **if** $Distances[i][j]$ *is undefined* **then**
18                         $Distance \leftarrow FindDistanceBetweenDataPoints(x^{(i)}, x^{(j)})$
19                         $Distances[i][j] \leftarrow Distance$
20                         $Distances[j][i] \leftarrow Distance$
21                     **if** $Distances[i][j] \leq \varepsilon_d$ **then**
22                         **if** $Labels[j][l] = 1$ **then**
23                             $Effects[i][l] \leftarrow Effects[i][l] + \frac{\beta}{(Distances[i][j])^\alpha}$
24                         **else**
25                             $Effects[i][l] \leftarrow Effects[i][l] - \frac{\beta}{(Distances[i][j])^\alpha}$
26                         **end**
27                 **end**
28         **end**
29     **end**
30 $boundary\_min, boundary\_max \leftarrow CalculateIQRBoundaries(LF\_stats, Effects, \xi)$
31 **for** $l = 1, 2, \ldots, m$ **do**
32     **for** $i = 1, 2, \ldots, |X|$ **do**
33         **if** $Labels[i][l] == -1$ **then**
34             **if** $Effects[i][l] < boundary\_min$ **then**
35                 $Labels[i][l] \leftarrow 0$
36             **if** $Effects[i][l] > boundary\_max$ **then**
37                 $Labels[i][l] \leftarrow 1$
38     **end**
39 **end**
40 **return** $Labels$

---

Alg. 1 shows the pseudocode of a similarity-based heuristic algorithm for reinforced labeling. Given $m$ LFs and the unlabeled dataset $X$ of size $|X|$, the algorithm outputs the augmented labeling matrix. The listed gravity parameters $\alpha := 1, \beta := 1$ are constants used for all datasets in the experimental study. The distance threshold $\varepsilon_d$ is an optional parameter to optimize the computation or to remove outliers, whereas it is not always used in the experimental study ($\varepsilon_d$ is set to a high number). $Labels, Effects$ are 2D arrays, where rows represent the index $i$ of the data points ($x^{(i)}$)

---

**Algorithm 2:** $CalculateIQRBoundaries(LF\_stats, Effects, \xi)$

---

**Input:** $LFs\_stats = \langle LFs\_coverage, LFs\_overlaps, LFs\_conflicts \rangle$ where
$\quad\quad LFs\_coverage = \langle LF_1\_coverage, LF_2\_coverage, \ldots, LF_m\_coverage \rangle$ and
$\quad\quad$ similarly $LFs\_overlaps$ and $LFs\_overlaps$; $Effects$: Array of aggregated effects,
$\quad\quad \xi \leftarrow 0.35$
**Output:** $boundary\_min, boundary\_max$

1   $sum_{coverage} \leftarrow 0$
2   $sum_{overlaps} \leftarrow 0$
3   $sum_{conflicts} \leftarrow 0$
4   **for** $l = 1, 2, \ldots, m$ **do**
5      $sum_{coverage} \leftarrow sum_{coverage} + LF_l\_coverage$
6      $sum_{overlaps} \leftarrow sum_{overlaps} + LF_l\_overlaps$
7      $sum_{conflicts} \leftarrow sum_{conflicts} + LF_l\_conflicts$
8   **end**
9   $h_{iqr} \leftarrow \xi * sum_{coverage} * sum_{overlaps} * sum_{conflicts}$
10   $q_1 \leftarrow calculatePercentile(Effects, 25)$
11   $q_3 \leftarrow calculatePercentile(Effects, 75)$
12   $iqr \leftarrow q_3 - q_1$
13   $boundary\_min \leftarrow q_1 - iqr * h_{iqr}$
14   $boundary\_max \leftarrow q_3 + iqr * h_{iqr}$
15   **return** $boundary\_min, boundary\_max$

---

and columns represent the index $l$ of the LFs ($LF_l$). $Distances$ is a 2D array representing the distance between any two data points. For instance, $Distances[i][j]$ represents the distance between $x^{(i)}$ and $x^{(j)}$.

The $-1$ values that represent abstains of LFs are updated based on their similarity given by the $FindDistanceBetweeDataPoints$ function. This function can be implemented using various distance metrics such as Euclidean, Cosine, or Mahalanobis distances. Lastly, the updated $Labels$ array represents the augmented labeling matrix that can be given to a generative process such as Snorkel's generative model.

Alg. 2 shows the pseudocode of the $CalculateIQRBoundaries$ used for the RL implementation. The heuristic in Alg. 2 leverages three LF statistics, that are LF coverage, overlaps, and conflicts, to calculate the boundaries that are effectively the aggregated effect threshold of RL.

## A.4 ADDITIONAL EXPERIMENTAL INSIGHTS

In addition to the experimental study, we implement and test a hybrid approach of leveraging labels of LFs as well as strong supervision through a gold dataset. We call this approach the *generative neural network* for data programming (GNN). In GNN, the label outputs of LFs and data features are fed to a simple neural network (NN) along with the labels. The NN model contains two hidden layers (# nodes 12, 8) with ReLU activation function, and it uses Adam optimizer. The output layer has the sigmoid activation function. The NN model is used in different stages of the pipeline. First, GNN replaces the generative part using labels of Snorkel (Sn+GNN+<endmodel>) or RL (RL+GNN+<endmodel>). Then, the outputs of GNN are fed to an end classifier as usual. Second, the GNN itself serves as the end classifier as well as the generative model (Sn+GNN or RL+GNN). We applied the GNN model to both red wine and white wine datasets as these datasets have the available ground truth data to create gold datasets.

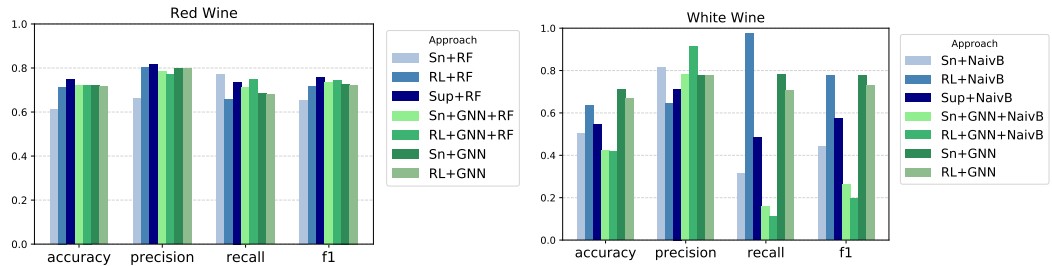

Figure 6: Experimental results of the red wine and white wine datasets for different approaches: Sn, RL, Sup., Sn+GNN, RL+GNN, Sn+GNN+<end_model> and RL+GNN+<end_model>.

Fig. 6 shows results of the red and white wine datasets in more detail, including the hybrid GNN approach using RF and NaivB end models, respectively. The results of the following 7 approaches are listed in order: *Sn+<endmodel>, RL+<endmodel>, Sup+<endmodel>, Sn+GNN+<endmodel>, RL+GNN+<endmodel, Sn+GNN, RL+GNN>*. We observe that RL+RF outperforms the Sn+RF benchmark for the white wine dataset by +13 points accuracy and +34 points F1 and even provides better results than Sup (RF). For the red wine dataset, RL+NaivB outperforms Sn+NaivB by +10 points in accuracy and +7 points in F1 score. Moreover, although approaches such as Sup or GNN leverage ground truth labels in their training, outcomes of RL are competitive for the red wine dataset, whereas RL outperforms Sup (NaivB), Sn+GNN+NaivB, and RL+GNN+NaivB for the white wine dataset (see Fig. 6-right).

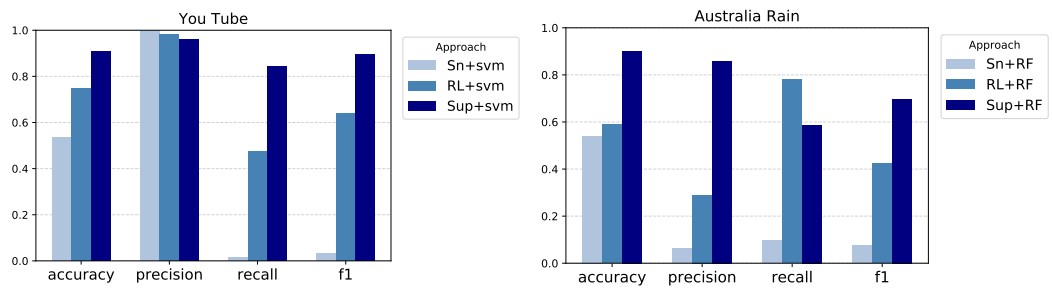

Figure 7: Experimental results for the YouTube comments and Australia rain datasets testing different approaches: Snorkel (Sn), Reinforced Labeling (RL), Supervised learning (Sup.).

Fig. 7 shows the bar graph for the results of the YouTube comments and Australia Rain datasets (also listed in Table 1) in terms of the four metrics: Accuracy, precision, recall, and F1. The results of the following approaches are listed in order: *Sn+<endmodel>, RL+<endmodel>, Sup+<endmodel>*. We observe that RL+svm outperforms the Sn+svm benchmark for the YouTube dataset by +21 points

accuracy and +61 points F1. For the Australia Rain dataset, RL+RF outperforms Sn+RF by +5 points in accuracy and +34 points in F1 score.

A.5   LABELING FUNCTIONS

We use the Snorkel library to implement LFs and encode domain knowledge programmatically. The LFs are implemented using the interactive and user-friendly features of the Snorkel framework, such as providing LF statistics and allowing high-level definitions of LFs.

As in almost all the previous data programming studies, we do not follow a certain scheme or strict guidance on implementing LFs but rather rely on the best effort based on some understanding of the datasets by visualizing the features and interactively checking the LF coverages, overlaps, and conflicts. In general, the assumption is that the developers would make the best effort to write LFs.

The Australia Weather dataset is used to build a model that predicts for any given day if it will rain the day after[4]. LFs in Listing 1 use weather data features to label data points as GOOD (1) or BAD (0) based on the features such as Humidity, Rain Today, Temperature at 9am, Humidity at 9am, and pressure at 3pm.

```
from snorkel.labeling import labeling_function
ABSTAIN = -1
NO_RAIN = 0
RAIN = 1

@labeling_function()
def check_humidity3pm(x):
    if x.Humidity3pm is None:
        return ABSTAIN
    elif x.Humidity3pm>0.75:
        return RAIN
    elif x.Humidity3pm<0.15:
        return NO_RAIN
    else:
        return ABSTAIN

@labeling_function()
def check_rain_today(x):
    if x.RainToday is None:
        return ABSTAIN
    elif x.RainToday==1:
        return RAIN
    else:
        return ABSTAIN

@labeling_function()
def check_temp9am(x):
    if x.Temp9am is None:
        return ABSTAIN
    elif x.Temp9am>0.60:
        return RAIN
    else:
        return ABSTAIN

@labeling_function()
def check_rainfall(x):
    if x.Rainfall is None:
        return ABSTAIN
    elif x.Rainfall>0.60:
        return RAIN
    else:
        return ABSTAIN

@labeling_function()
def check_humidity9am(x):
    if x.Humidity9am is None:
        return ABSTAIN
```

---

[4]https://www.kaggle.com/jsphyg/weather-dataset-rattle-package

```
48      elif x.Humidity9am>0.90:
49          return NO_RAIN
50      elif x.Humidity9am<0.20:
51          return RAIN
52      else:
53          return ABSTAIN
54
55  @labeling_function()
56  def check_pressure3pm(x):
57      if x.Pressure3pm is None:
58          return ABSTAIN
59      elif x.Pressure3pm<0.05:
60          return RAIN
61      elif x.Pressure3pm>0.70:
62          return NO_RAIN
63      else:
64          return ABSTAIN
```

Listing 1: Australia rain labeling functions

The wine datasets have various data features such as alcohol, sulfates, citric acid levels, etc. Listing 2 and 3 include the LFs implemented for wine quality classification for the red wine and white wine datasets, respectively. LFs labels data points as GOOD (1) or BAD (0) quality wine.

```
1  from snorkel.labeling import labeling_function
2  ABSTAIN = -1
3  BAD = 0
4  GOOD = 1
5
6  @labeling_function()
7  def check_alcohol(x):
8      if x.alcohol is None:
9          return ABSTAIN
10     elif x.alcohol>0.75:
11         return GOOD
12     elif x.alcohol<0.15:
13         return BAD
14     else:
15         return ABSTAIN
16
17 @labeling_function()
18 def check_sulphate(x):
19     if x.sulphates is None:
20         return ABSTAIN
21     elif x.sulphates>0.3:
22         return GOOD
23     else:
24         return ABSTAIN
25
26 @labeling_function()
27 def check_citric(x):
28     if x.acidity_citric is None:
29         return ABSTAIN
30     elif x.acidity_citric>0.7:
31         return GOOD
32     else:
33         return ABSTAIN
```

Listing 2: Red wine labeling functions

```
1  from snorkel.labeling import labeling_function
2  ABSTAIN = -1
3  BAD = 0
4  GOOD = 1
5
```

```
6  @labeling_function()
7  def check_alcohol(x):
8      if x.alcohol is None:
9          return ABSTAIN
10     elif x.alcohol>0.75:
11         return GOOD
12     elif x.alcohol<0.15:
13         return BAD
14     else:
15         return ABSTAIN
16
17 @labeling_function()
18 def check_sulphate(x):
19     if x.sulphates is None:
20         return ABSTAIN
21     elif x.sulphates>0.3:
22         return GOOD
23     else:
24         return ABSTAIN
25
26 @labeling_function()
27 def check_citric(x):
28     if x.acidity_citric is None:
29         return ABSTAIN
30     elif x.acidity_citric>0.7:
31         return GOOD
32     else:
33         return ABSTAIN
```

Listing 3: White wine labeling functions

For the YouTube dataset, we implemented two LFs that search for the exact string *"check out"* or *"check"* in the text. Other than these two additional LFs, the LFs in the Snorkel tutorial [Snorkel (2021)] named "textblob_subjectivity", "keyword_subscribe", "has_person_nlp" are used in the experiments of Table 1. The rest of LFs available from the Snorkel tutorial are used for the experiments in Fig. 5.

```
1  from snorkel.labeling import labeling_function
2  ABSTAIN = -1
3  HAM = 0
4  SPAM = 1
5
6  @labeling_function()
7  def check(x):
8      return SPAM if "check" in x.text.lower() else ABSTAIN
9
10 @labeling_function()
11 def check_out(x):
12     return SPAM if "check out" in x.text.lower() else ABSTAIN
```

Listing 4: YouTube labeling functions

