# OpenReview forum: "Label Augmentation with Reinforced Labeling for Weak Supervision"
_ICLR.cc/2022/Conference — ICLR 2022 Submitted_

### Official Review · Reviewer_5fTq · 2021-10-21

**Correctness:** 3
**Technical Novelty And Significance:** 2
**Empirical Novelty And Significance:** 2
**Recommendation:** 5
**Confidence:** 4

**Main Review:**

**Strengths**
- This paper focuses on a practical problem. The motivation is good.
- The experimental results are promising. In almost all cases, we can see that the improvement on F1-Score is clear.

**Weaknesses/Questions**
- The organization of this paper is a bit confused. I understand the core idea and algorithm flow of this paper after I read it a few times.
- The writing is a bit illogical and makes readers confused. For example, this paper tends to augment the outputs of labeling functions for better generalization. Perhaps, it is expected to see that the intuition about why the proposed method works and how it handles the issues. However, the explanation is a bit incomprehensible.
- Figures 2 and 3 are not informative. Also, the explanations are not sufficient.
- In Equation 1, there are some hyper-parameters. How to control them in experiments?
- I am confused about the auto-adjustment of $\epsilon$, which is dependent on the dataset. However, it seems that we still need to determine $\xi$ artificially to control $\epsilon$?


**Summary Of The Paper:**

This paper tackles the problem of designing data programming, which is a practical approach in weak supervision. The authors state that prior effects neglect to utilize data features during the generative process, and therefore have suboptimal performance. Based on this, this paper proposes to exploit the sample similarities to augment the outputs of labeling functions. Empirical evaluations are provided to verify the effectiveness of the proposed method.

**Summary Of The Review:**

This paper tackles a practical problem. The idea is relatively novel. However, the writing of this paper needs to be improved to reach the acceptance line. The current version is hard to follow. Therefore, my score for this paper is negative.

---

> ### Author Response · Authors · 2021-11-19
> **Response to comments of Reviewer 5fTq**
>
> We thank the reviewer for their effort and comments on the paper.
>
> We understand that the organization of the paper might not be easy to follow. We consider adding additional explanations to Fig. 2 and Fig. 3. We believe that the reviewer understood the major points of this work.
>
> We agree with the reviewer on the parameter setting, the threshold parameter should be set empirically, whereas we believe existing statistics from the dataset and labeling functions (e.g., labeling coverage, overlap, conflicts) can be useful to automate this process. We show the importance of those parameters and an approach to make auto-adjustment of them. However, this is a limitation of the RL approach implementation as pointed out in section 3.2. This is a point for future work or for having a completely new approach. Nevertheless, RL demonstrates that the usage of data features during the generative process is beneficial.

---

### Official Review · Reviewer_eeud · 2021-11-02

**Correctness:** 3
**Technical Novelty And Significance:** 3
**Empirical Novelty And Significance:** 1
**Recommendation:** 3
**Confidence:** 4

**Main Review:**

Strength:
1.	The research problem of weak supervision is very important and worth studying.
2.	The proposed approach for data programming can improve the performance of weak supervision.
3.	The proposed approach alleviated two limitations of existing data programming methods: (1) coarse information, and (2) lack of generalization compared to existing Snorkel framework by taking data features into considerations.
Weakness/concerns:
1.	The technical novelty of this work is limited as the proposed reinforced labeling is similar to KNN with pseudo-labels. How these two methods differ in underlying motivation, method design, and experiments is not properly explored.
2.	The term ‘generative’ would seriously confuse readers as it was not referring to data generation but to label generation, it would be better if the authors could use ‘label generative’ instead of ‘generative’.
3.	The experiments are not convincing. What is the supervised learning method the authors used to compete with their weak supervision method? Is there any improvement for the supervision performance if the authors incorporate their weak supervision approach (or pseudo-labels) into a supervised setting?
4.	The labeling functions seems to be rule-based functions that maps the features to the pseudo labels. I’m not sure how this can bring in new information to the downstream classifier when it is powerful enough. How do you avoid the error propagation problem? Also, if the pseudo labels are good enough, why do you need a downstream classifier as the pseudo labels are already the upper bound of the classifier?


**Summary Of The Paper:**

The authors proposed a new approach called reinforced labeling to enhance the data programming weak-supervision approach. Labeling functions in data programming can only produce pseudo-labels for part of the dataset, and thus the authors propose to use similarity between two data features to find the lacking pseudo-labels, and thus enhance the existing data programming methods.

**Summary Of The Review:**

The work may be interesting for the automatic label annotation area, but the idea novelty is a little trivial and experimental results are not convincing.

---

> ### Author Response · Authors · 2021-11-19
> **Response to comments of Reviewer eeud**
>
> We thank the reviewer for their effort and comments on the paper.
>
> #4-response: To the best of our knowledge, this is the first study that considers adjusting label augmentation of the labeling function outcomes, as opposed to labeling propagation. We are not sure which study is referred in comment #4, but by first look they seem a different approach.
>
> #5-response: We agree with the suggestion by the reviewer to change it and make it clearer. The term generative is mainly used in the weak-supervision studies in recent years (generative model and discriminative model); we wanted to used that terminology to avoid confusions.
>
> #6-response: In this study, we stick on using no ground-truth supervision and stay in the weak supervision paradigm. It would be interesting to explore the hybrid model (supervised and weak-supervised) in the context of active learning. In the experiments, supervised learning (Sup) is used as a baseline comparison. As mentioned into section 3.1, Sup baseline uses the same end-model of RL and Snorkel with 30% ground-truth training dataset.
>
> #7-response: To avoid the error propagation we consider adjusting the IQR factor and follow a more conservative approach especially for high coverage cases. On the other hand, existing programmatic labeling and this paper both assume that the most (if not all) labeling functions would have certain accuracies that are better than a random guess.

---

### Official Review · Reviewer_HUob · 2021-11-02

**Correctness:** 2
**Technical Novelty And Significance:** 2
**Empirical Novelty And Significance:** 1
**Recommendation:** 3
**Confidence:** 5

**Main Review:**

This paper discusses an interesting method and is well-written and clear to follow. However, there appear to be major problems with the evaluation, and the method is not novel.

Evaluation:
The authors only compare to one baseline, Snorkel, and report significantly worse performance on benchmark datasets than reported elsewhere in the literature. Consider the YouTube dataset -- this is a standard weak supervision dataset and is used as a tutorial for the open-source Snorkel library.

In the tutorial itself (https://www.snorkel.org/use-cases/01-spam-tutorial), Snorkel achieves an accuracy of 94.4%. However, the authors report an accuracy of 54%, with an F1 of 3% in Table 1. This suggests that the authors did not run Snorkel correctly. In fact, the authors' reported performance on the YouTube dataset (75% accuracy) significantly underperforms the tutorial notebook. This discrepancy is because the authors did not use all the labeling functions in the YouTube dataset. This is a non-standard evaluation procedure, and makes the results hard to compare and believe.

In future iterations, the authors should also consider comparing against the baselines in the WRENCH benchmark (https://arxiv.org/pdf/2109.11377.pdf) as well, to ensure that their evaluation is fair and standard. (This paper was released <1 month before submission, so I don't count it against them -- but it's a useful datapoint nonetheless).

I am less familiar with the wine and weather datasets, since I have not seen them in other WS datasets, but performance is suspiciously low for those as well (F1 of 8% for Snorkel on Weather, which suggests that Snorkel was not tuned correctly).

Novelty:
The method itself is almost identical to that in Chen et al 2020 (https://arxiv.org/abs/2006.15168). The authors cite this paper but do not compare against it. Chen et al also focus on improving labeling function coverage by replacing abstentions based on similarity in feature space to other points. Both methods feature a hard threshold after which they no longer transfer labeling function labels. The one new point is that this paper discusses a method for automatically adjusting the threshold (section 3.2), whereas Chen et al tuned the thresholds manually. I suggest anchoring more strongly on this contribution in the future.

**Summary Of The Paper:**

This paper describes a method to improve the coverage of labeling functions in weak supervision. Labeling functions in weak supervision can abstain if they do not vote on a particular point. The method replaces abstentions with a gravitation-based method based on the other points that the labeling function has voted on. The method uses similarities between features of points for the gravitation-based method. The authors claim performance benefits over Snorkel.

**Summary Of The Review:**

This paper is clearly-written, but it has significant flaws in its evaluation, and the method is only marginally novel. Therefore I recommend reject.

---

> ### Author Response · Authors · 2021-11-19
> **Response to comments of Reviewer HUob**
>
> We thank the reviewer for their effort and comments on the paper.
>
> In the experiments, we present the situations where the existing model does not perform well due to various reasons (e.g., limited labeling functions) and show the advantage of the label augmentation in those scenarios. The Snorkel tutorial applied on the YouTube dataset works with very high performances when all the given LFs are used for creating the labeling matrix. However, we wanted to show that in case of small coverage of LFs (such as very few LFs) our proposed approach can still bring to good performances. Thus, we have run the Snorkel notebook and limited the use of all LFs. Indeed, we assume scenarios where a data scientist has a dataset but no LFs. Thus, the data scientist might implement the bare minimum set of LFs. Nevertheless, in Figure 5 we show that by increasing the number of LFs Snorkel catches up with the performance reaching higher accuracy while our RL method might add too much noise. For such reason, we have study a way to reduce the effects of RL with an auto-adjustment method.
>
> For future iterations, we plan to apply Wrench framework as next step. We thank the reviewer to suggest this.
>
> We believe both approaches by Chen et al. and this paper promise improvements on the existing data programming as they both consider the data features on the early phase of the pipeline which is missing in overwhelming majority of the studies. In this paper we identify the key problem of the existing data programming pipeline and propose a more general label augmentation approach that is useful to benefit features from every type of data (not focusing on NLP) and without the need of pretrained ML models. Lastly, our model can conservatively label the data points and it does not have to assign every data point a label.

---

> > ### Comment · Reviewer_HUob · 2021-11-29
> > **Rebuttal Response**
> >
> > Thank you for your response. I agree that applying your method to the datasets in the Wrench benchmark in future iterations would improve the paper, but as-is I will be keeping my score.
> >
> > I agree that the justification of looking at the setting with limited LF's is an interesting one, but I would suggest comparing against more baselines for this setting as well. For example, Snuba (http://www.vldb.org/pvldb/vol12/p223-varma.pdf) automatically generates labeling functions and could be an interesting baseline for this setting. Boecking et al (https://arxiv.org/abs/2012.06046) also look at this problem from the perspective of someone writing new LF's for the first time. I'll also point out Chen et al again -- they identify the same coverage problem as you do, and see lift even with all of the Snorkel LF's (additionally, they don't focus on NLP, and they do not have to assign every data point a label as you claim).

---

### Official Review · Reviewer_s9uq · 2021-11-03

**Correctness:** 2
**Technical Novelty And Significance:** 2
**Empirical Novelty And Significance:** 2
**Recommendation:** 3
**Confidence:** 5

**Main Review:**

Strengths:
- (S1) The proposed approach tackles a relevant area of the space with appealing top-level motivation and intuition.

Weaknesses:
- (W1) Lack of adequate comparisons to related work: There are many weak supervision approaches that have been proposed since the original Snorkel paper referenced and compared to in the experimental results; some of which are referenced in the paper, but never actually compared to.  These more recent papers address many of the proposed issues- incorporating data features, boosting generalization with joint learning, etc.  Given the motivation of the paper, and claims in the paper that the proposed approach is better than these other approaches cited, it seems to be a major gap that no comparisons are made to these other methods in the experiments.
- (W1a) Additionally, this work is very similar to classic label propagation techniques in semi-supervised learning, and should have been compared to these as well.

- (W2) Cherry-picked experimental results: There are several aspects of the experiments that raise significant questions as to the protocol with which they were conducted, and the representativeness of the experimental configurations; for example:
  * (W2a) In the main results table (Table 1), Snorkel is reported as having an average F1 of 0.03 on the YouTube dataset (this is a score so low as to be actively difficult to achieve in most settings!).  The YouTube dataset, as noted in the paper and appendix explicitly, is *from the Snorkel OSS intro tutorial, in which Snorkel achieves a 94+% accuracy* (see snorkel.org/use-cases/01-spam-tutorial).  Yet somehow, the results here are reported as 0.03 F1; according to the appendix, because of an arbitrary subselection of LFs.  In fact, in Fig.5(a) you can see that with more LFs, the score goes up significantly and either matches or outperforms the proposed approach (see next point for more here).  It therefore seems extremely misleading to have presented the scores as presented in Table 1 as main results.  If the objective was to show that in certain low-LF situations, the proposed approach can offer benefits, this should have been messaged clearly upfront when reporting main results.
  * (W2b) The proposed approach is (like most/any clustering-based approaches) highly dependent on a threshold parameter, the so called "IQR factor".  In the main results, this is apparently "chosen by a data scientist".  This raises significant questions as to experimental protocol

- (W3) Lack of sufficient ablation and/or formal motivation for proposed approach: The proposed approach is presented in a way that makes it seem fairly arbitrary compared to existing approaches for e.g. weak supervision, clustering, label propagation, etc.  However, minimal ablations of the approach are performed (just the threshold factor and the distance metric) to actually motivate or defend the heuristic approach taken.  This makes it hard to gain insight from the paper about why this approach should work and why it should work better than other approaches- esp. given the lack of other methods compared to.

**Summary Of The Paper:**

This paper proposes an extension to the data programming / weak supervision method of Ratner et. al. 2016, in which heuristic functions called labeling functions (LF) are used to label training data.  In the proposed extension, called "reinforced labeling", points that are close to those labeled by a labeling function in feature space (as determined by a heuristic "gravitation" algorithm) are also labeled, thereby augmenting the set of programmatically labeled examples.

**Summary Of The Review:**

This paper lacks sufficient comparison to other more contemporary weak supervision work, lacks sufficient motivation and/or ablation w.r.t. the heuristic design choices taken, and has potential issues with experimental protocol.

---

> ### Author Response · Authors · 2021-11-19
> **Response to comments of Reviewer s9uq**
>
> We thank the reviewer for their effort and comments on the paper.
>
> W1-response: We believe our approach would be complementary to the existing solutions, in other words, it can be used along with the other solutions. In the experimental study we show its use on top of the existing Snorkel and showcased significant performance benefits. Thus, our approach can be used in combination with other weak supervised approach.
>
> W1a-response: In our work, we mainly identified a limitation of the known data programming pipeline that neglects to utilize data features during the generative process. Our reinforced labeling approach demonstrates that what we have identified is a limitation. Techniques for label propagation approach can be also beneficial, our proposed approach considers propagation on the labeling matrix (LF outputs).
>
> W2-response: The results we highlight are the ones where our approach is working better. This helps on showing when RL is good to be applied that is when LFs are few or with a cumulative small coverage. Nevertheless, in Figure 5 we show that by increasing the number of LFs Snorkel catches up with the performance reaching higher accuracy while our RL method might add too much noise. For such reason, we have study a method to reduce the effects of RL with an auto-adjustment method.
>
> W2a-response: The results we present is not to discredit Snorkel but to show that in case when there is small coverage of LFs our approach can bring the pipeline to work. We stated this into the introduction already: “Label augmentation extends the data programming to new scenarios, such as when LFs have low coverage, domain experts can implement only a limited number of LFs, or LFs outcome result in a sparse labeling matrix”. Indeed, we assume scenarios where a data scientist has a dataset but no LFs. Thus, he might implement the bare minimum set of LFs. Scenarios well engineered such as in the Snorkel tutorial and with many LFs with good coverage are shown in Figure 5, where Snorkel beats RL.
>
> W2b-response: In the paper, we discuss how the IQR factor can be configured automatically. We pointed this out among the limitations and as future work.
>
> W3-repsonse: In the paper, we have shown the impact of RL by setting different values of IQR factor. Further, we show the impact of the distance metric on our approach. We also show in the appendix the results with different end-model (RL is beneficial in all of them). We believe that it is enough to show that using data features during the generative process is beneficial.

---

### Decision · Program_Chairs · 2022-01-20

**Decision:**

Reject

**Comment:**

The paper presents an approach to weak supervision to address the possibly low-coverage of rule-based labeling functions, by assigning similar labels to similar instances (where the similarity is computed in feature space).

The reviewers main concerns were the presentation, as well as the experimental protocol and results. Several directions for improvement have been identified by the reviewers and acknowledged by the authors, but in the current state of the submission the consensus is that the paper is not ready for publication.